# Learning to predict target location with turbulent odor plumes

**Nicola Rigolli[1,2,3,4]\*, Nicodemo Magnoli[1,3], Lorenzo Rosasco[5], Agnese Seminara[2,4]\***

[1]Department of Physics, University of Genova, Genova, Italy; [2]Institut de Physique de Nice, Université Côte d'Azur, Centre National de la Recherche Scientifique, Nice, France; [3]National Institute of Nuclear Physics, Genova, Italy; [4]MalGa, Department of Civil, Chemical and Environmental Engineering, University of Genoa, Genoa, Italy; [5]MaLGa, Department of computer science, bioengineering, robotics and systems engineering, University of Genova, Genova, Italy

**Abstract** Animal behavior and neural recordings show that the brain is able to measure both the intensity and the timing of odor encounters. However, whether intensity or timing of odor detections is more informative for olfactory-driven behavior is not understood. To tackle this question, we consider the problem of locating a target using the odor it releases. We ask whether the position of a target is best predicted by measures of timing *vs* intensity of its odor, sampled for a short period of time. To answer this question, we feed data from accurate numerical simulations of odor transport to machine learning algorithms that learn how to connect odor to target location. We find that both intensity and timing can separately predict target location even from a distance of several meters; however, their efficacy varies with the dilution of the odor in space. Thus, organisms that use olfaction from different ranges may have to switch among different modalities. This has implications on how the brain should represent odors as the target is approached. We demonstrate simple strategies to improve accuracy and robustness of the prediction by modifying odor sampling and appropriately combining distinct measures together. To test the predictions, animal behavior and odor representation should be monitored as the animal moves relative to the target, or in virtual conditions that mimic concentrated *vs* dilute environments.

**\*For correspondence:**
nicola.rigolli@edu.unige.it (NR);
agnese.seminara@unige.it (AS)

## Editor's evaluation

This paper explores the question of the optimum strategy for odor detection in a turbulent environment. The authors use high-resolution simulations of turbulent flow to investigate the transport and detection of odors advected by the flow, comparing machine learning strategies based on the temporal dynamics of the signal with those based on intensity. The work should be of interest to researchers working on a broad range of problems in sensation and navigation across scales.

## Introduction

Most macroscopic organisms detect odors in intermittent bursts, that may be separated by extended regions with no odor. Organisms leverage this complex dynamics efficiently for diverse tasks, including locating and identifying an odor source *Murlis et al., 1992*; *Mafra-Neto and Cardé, 1994*; *Vickers, 2000*; *Riffell et al., 2014*; *Ache et al., 2016*; *Ackels et al., 2021*. However, what are the most informative features of intermittent odor cues remains largely unclear. There are two broad classes of measures that quantify the dynamics of olfactory cues: those that depend on odor intensity including e.g. odor gradients in space or time, and those that do not depend on odor intensity but only on its timing, i.e. on whether the odor is on or off regardless of its concentration. To compute quantities that

depend on odor intensity, an accurate representation of the odor is needed. In contrast, measuring the timing of odor detection simply requires to mark at all times whether the odor is on or off, thus a binary switch is sufficient.

Behavioral evidence suggests that animals use both intensity and timing of odor encounters for olfactory navigation *Baker et al., 2018*. At close range, mammals appear to compare odor intensity either across nostrils or across sniffs *Catania, 2013*; *Gire et al., 2016*; *Findley et al., 2021*. On the other hand, mounting evidence suggests timing of odor detection also plays a key role for olfactory navigation *Ache et al., 2016*: moths respond to odor pulsed at specific frequencies *Riffell et al., 2014*; *Vickers et al., 2001*; fruit flies respond to timing since last odor detection *van Breugel and Dickinson, 2014*; *Demir et al., 2020*; lobsters and sharks compare odor arrival time across their paired olfactory organs and orient toward the side that detected the odor first *Basil and Atema, 1994*; *Gardiner and Atema, 2010*; many organisms will move upwind upon detection of an odor *Kennedy and Marsh, 1974*; *Murlis et al., 1992*; *Steck et al., 2012*.

Neural recordings upon stimulation with intermittent odor cues confirm that the brain of many animals is able to record information both about intensity (and its derivatives) as well as timing of odor encounters (most information comes from work on arthropods *Nagel and Wilson, 2011*; *Vickers et al., 2001*; *Brown et al., 2005*; *Gorur-Shandilya et al., 2017*; *Jacob et al., 2017*; *Riffell et al., 2014*, but see also *Parabucki et al., 2019*; *Lewis et al., 2021*). For example, when insects are presented with intermittent odor cues, information about intensity and timing is recorded in their antennal lobe (see e.g. *Vickers et al., 2001*; *Brown et al., 2005*). Odors that mimic natural intermittency elicit a response that preserves an accurate measure of timing in fruit flies and moths *Gorur-Shandilya et al., 2017*; *Jacob et al., 2017*. In lobsters, bursting olfactory neurons encode specifically for the time between successive odor encounters, see *Park et al., 2014*; *Park et al., 2016* and references therein. Interestingly, the neural activity varies considerably with the dynamics of the odor cues *Nagel and Wilson, 2011*; *Vickers et al., 2001*; *Lewis et al., 2021*, but how intermittency of an odor affects its neural representation is not well understood.

This evidence suggests animals are able to identify when they detect an odor as well as how intense it is; but whether they record and rely on both kinds of information is not understood. From a physical perspective, these two measures clearly provide information about source location. Indeed, we know from theoretical *Shraiman and Siggia, 2000*; *Falkovich et al., 2001*; *Celani et al., 2014* and experimental *Justus et al., 2002*; *Moore and Crimaldi, 2004* work that turbulence causes the odor to be distributed in highly intermittent patches separated by blanks with no odor. Both intensity and timing of these intermittent bursts vary depending on the location of the source *Celani et al., 2014*, as early recognized by *Atema, 1996*, thus can be used to infer source location or navigate to it *Vergassola et al., 2007*; *Schmuker et al., 2016*; *Boie et al., 2018*; *Leathers et al., 2020*; *Michaelis et al., 2020*.

Here, we ask what salient features of turbulent odor signals best predict the location of the odor source and specifically compare quantities related to intensity *vs* timing of odor encounters. We first compose a dataset of realistic odor fields at scales of several meters using accurate state-of-the-art fluid dynamics simulations. We then develop machine learning algorithms that predict source location based on these synthetic odor fields.

We find that measures of odor temporal dynamics based on a short memory span (down to about 1 s) hold information about source location. Close to the source or close to the substrate, measures of intensity predict distance better than measures of timing; but this ranking is reversed at further distance from the source or from the substrate. Pairing the two kinds of measure improves dramatically the quality of the prediction robustly across all datasets, whereas pairing two measures of intensity or two measures of timing is either useless or detrimental.

Our results demonstrate that timing and intensity are complementary attributes of odor dynamics and are most effective in more dilute and concentrated conditions respectively. These different conditions exist in different portions of space because odor gets transported, mixed and diluted by the fluid. As a result, the spatial range of operation of a living organism constrains the solutions it may evolve to make predictions with turbulent odors.

## Results

Odor cues at several meters from the source are often turbulent. *Figure 1a–c* and show snapshots of the velocity field and odor cues in space, resulting from direct numerical simulations of the turbulent

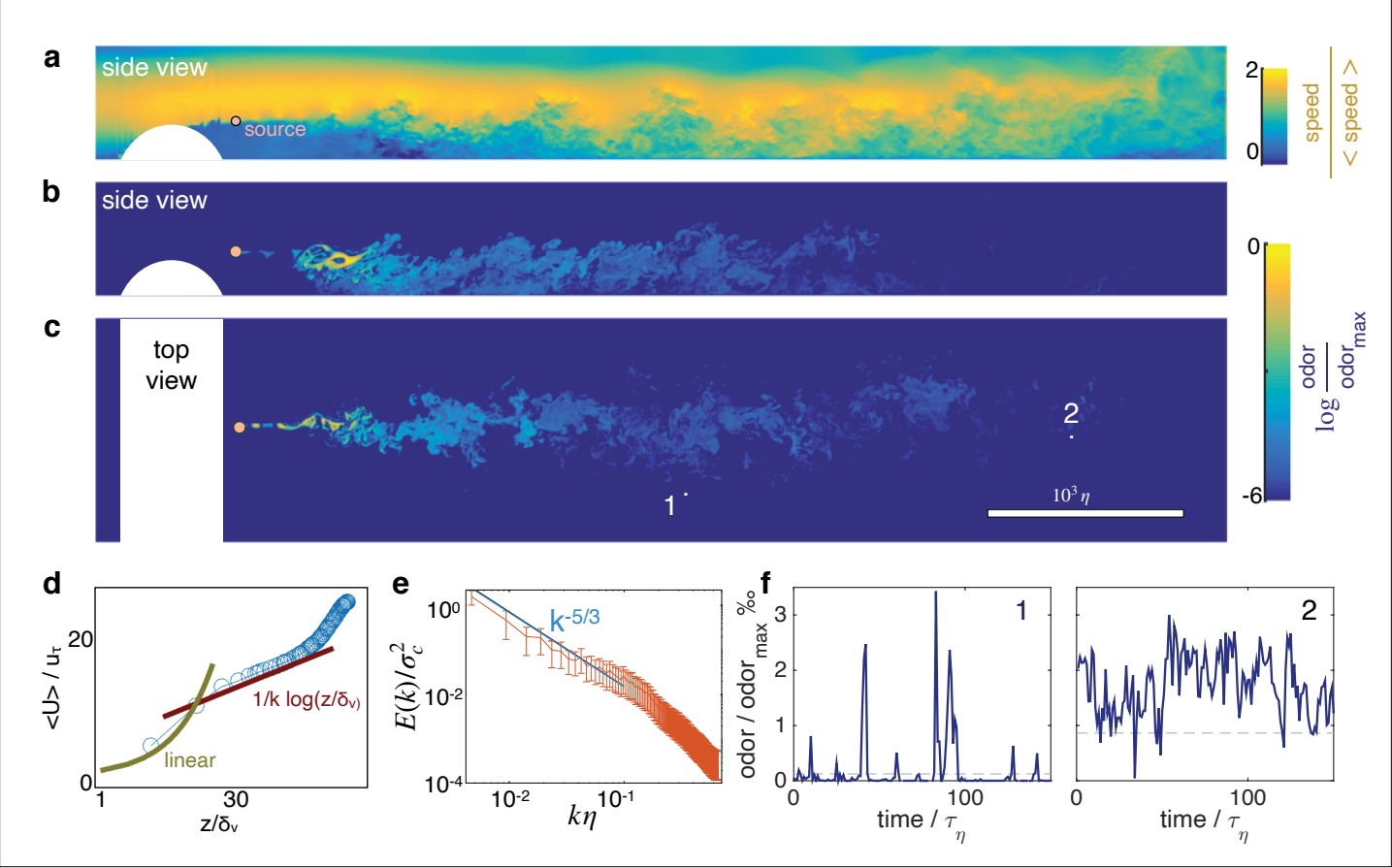

**Figure 1.** Turbulent odor cues are patchy and intermittent. Snapshot of streamwise velocity (**a**) in a vertical plain at mid channel; odor snapshot side view at mid channel (**b**) and top view at source height (**c**). White regions mark the cylindrical obstacle. Snapshots are obtained from direct numerical simulations of the Navier-Stokes equations and the equation for odor transport (see Materials and Methods and parameters summarized in *Table 1*). (**d**) The mean velocity profile follows the well known log law when $z_3^+ = z_3/\delta_\nu > 30$, where $\delta_\nu = \nu/u_\tau$ where $u_\tau$ is the friction velocity. (**e**) Two dimensional spectra of odor fluctuations $E(k) = \frac{d}{dk}(\int_{|\mathbf{k}_{xy}|<k} |\hat{c}(\mathbf{k}_{xy})|^2 d^2k)$ normalized with the scalar variance $\sigma_c^2$; $\hat{c}(\mathbf{k}_{x,y})$ is the 2D Fourier transform of the scalar concentration at source height; the integral of the spectra is the scalar variance. Wavenumbers are nondimensionalized with the inverse Kolmogorov scale $\eta^{-1}$. Error bars show standard deviation calculated over N = 420 points. (**f**) Typical time courses of the odor cues at locations labeled with 1 and 2 in c, visualizing noise and sparsity, particularly at location 1.

The online version of this article includes the following video for figure 1:

**Figure 1—video 1.** Direct numerical simulations of the turbulent flow in a channel of length L, width W and height H evolving in time.
https://elifesciences.org/articles/72196/figures#fig1video1

flow in a channel of length L, width W and height H (also see *Figure 1—video 1*). Air flows from left to right at a mean speed $U_b$ and hits a cylindrical obstacle that generates turbulence. The height of the obstacle is $H/4$ and tunes the intensity of turbulent fluctuations relative to the mean velocity. To characterize the flow we show in *Figure 1d* that the mean velocity profile, for $z^+ \gtrsim 30$, follows the law of the wall $\frac{\langle U \rangle}{u_\tau} = \frac{1}{\kappa} \ln z^+ + B$, where $\kappa$ and $B$ are constants and $z^+ = \frac{z}{\delta_\nu}$, recovering classical statistics for channel turbulence *Pope, 1984*. The odor field is emitted from a concentrated source downstream from the obstacle; it develops as a meandering filament that fluctuates as it travels downstream and soon breaks into discrete pockets of odor (whiffs) separated by odor-less stretches (blanks) (*Figure 1b–c and f*). The Kolomogorov scaling for the spectra of odor fluctuations $k^{-5/3}$ holds for $k\eta \lesssim 0.1$ (see *Figure 1e*), consistent with previous experimental results in channel flow (see e.g. *Saddoughi and Veeravalli, 1994*). Typical time courses of the odor are shown in *Figure 1f*. Note that depending on the sampling location, odor may be more or less sparse (compare for example *Figure 1f* left and right). All parameters and methods are summarized respectively in *Table 1* and in Materials and Methods.

**Table 1.** Parameters of the simulation.

Length $L$, width $W$, height $H$ of the computational domain; horizontal speed along the centerline $U$; mean horizontal speed $U_b = \langle u \rangle$; kinematic viscosity $\nu$; diffusivity $\kappa_c$; Kolmogorov length scale $\eta = (\nu^3/\epsilon)^{1/4}$ where $\epsilon$ is the energy dissipation rate; mean size of gridcell $\Delta x$; Kolmogorov timescale $\tau_\eta = \eta^2/\nu$; energy dissipation rate $\epsilon = \nu/2\langle(\partial u_i/\partial x_j + \partial u_j/\partial x_i)^2\rangle$; Taylor microscale $\lambda = \sqrt{\langle u^2 \rangle/\langle(\partial u/\partial x)^2\rangle}$; wall lengthscale $y^+ = \nu/u_\tau$ where the friction velocity is $u_\tau = \sqrt{\tau/\rho}$ and the wall stress is $\tau = \rho\nu du/dz|_{z=0}$; Reynolds number $Re = U(H/2)/\nu$ based on the centerline speed $U$ and half height; Reynolds number $Re_\lambda = U\lambda/\nu$ based on the centerline speed and the Taylor microscale $\lambda$; Schmidt number (Sc = $\nu/\kappa_c$ = Pe/Re); magnitude of velocity fluctuations $u'$ relative to the centerline speed; large eddy turnover time $T = H/2u'$. First row reports results in non dimensional units; second and third rows correspond to dimensional parameters in air and water assuming the velocity of the centerline is 50 cm/s in air and 12 cm/s in water.

|  | $L$ | $W$ | $H$ | $U$ | $U_b$ | $\nu$ | $\kappa_c$ | $\eta$ | $\Delta x$ |
|---|---|---|---|---|---|---|---|---|---|
|  | 40 | 8 | 4 | 32 | 23 | 1/250 | 1/250 | 0.006 | 0.025 |
| air | 9.50 m | 1.90 m | 0.96 m | 50 cm/s | 36 cm/s | $1.5 10^{-5}$ m²/s | $1.5 10^{-5}$ m²/s | 0.15 cm | 0.6 cm |
| water | 2.66 m | 0.53 m | 0.27 m | 12 cm/s | 8.6 cm/s | $10^{-6}$ m²/s | $10^{-6}$ m²/s | 0.04 cm | 0.2 cm |
|  | $\tau_\eta$ | $\epsilon$ | $\lambda$ | $y^+$ | Re | $Re_\lambda$ | Sc | $u'/U$ | $T$ |
|  | 0.01 | 39 | 0.17 | 0.0035 | 16000 | 1360 | 1 | 11% | $64\,\tau_\eta$ |
| air | 0.15 s | 6.3e-4 m²/s³ | 4 cm | 0.09 cm |  |  |  |  |  |
| water | 0.18 s | 3e-5 m²/s³ | 1 cm | 0.02 cm |  |  |  |  |  |

Do odor cues bear information about source location meters away from the source? To answer this question, we develop supervised machine learning algorithms that learn the relationship between the input (odor) and the distance from the source (output) from a large dataset of examples. In order to dissect what are the best predictors of source location and how ranking depends on the statistics of the odor, we need to detail more specifically the input and output of the algorithm.

To design the input we start with the odor concentration field $c(\mathbf{z}, t)$ which varies stochastically in space and time as a result of turbulent transport. Here, $\mathbf{z} = (z_1, z_2, z_3)$ is a location in the three dimensional space and $t$ is time. We focus on a plane at a fixed height, and consider the conical region where odor can be detected, the 'cone of detection' (*Figure 2a*). We first compose time series of the odor field; each time series is indicated with $\mathbf{c}_i$ and consists of the odor sampled at $M$ equally spaced times with frequency $\omega$ at a discrete location $\mathbf{z}_i$ within the cone of detection. Thus, each time series is a vector $\mathbf{c}_i = (c(\mathbf{z}_i, t_i), ..., c(\mathbf{z}_i, t_{i+M}))$, where $t_{i+M} - t_i = M/\omega$ is the temporal span of the time series, or memory. From each time series, $\mathbf{c}_i$ we calculate five features $x_i^1, ..., x_i^5$, where $x_i^1$ is the temporal average of the concentration during whiffs in the time series $\mathbf{c}_i$; $x_i^2$ is its average slope (time derivative of odor upon detection, averaged across whiffs within $\mathbf{c}_i$); $x_i^3$ is the average duration of blanks (stretches of time when odor is below detection within $\mathbf{c}_i$); $x_i^4$ is the average duration of whiffs (stretches of time when odor is above threshold within $\mathbf{c}_i$); and $x_i^5$ is the intermittency factor (the fraction of time the time series $\mathbf{c}_i$ is above threshold). The detection threshold is defined adaptively as discussed in Materials and methods and *Figure 2—figure supplement 1*. Features $x^1$ and $x^2$ depend explicitly on odor concentration, whereas features $x^3$, $x^4$ and $x^5$ only depend on when the odor is on or off, but not on its intensity. To remark this difference, we refer to $x^1$ and $x^2$ as intensity features, and $x^3$, $x^4$ and $x^5$ as timing features. Our input $\mathbf{x}_i = (x_i^1, ..., x_i^d)$ is composed of d-dimensional vectors of features and we will focus on $d = 1, 2, 5$. We seek to infer distance from the source, thus our output $y$ is the coordinate of the sampling point $\mathbf{z}$ in the downwind direction, i.e. $y = z_1$, with the source placed at the origin (see sketch in *Figure 2a*). We refer to the figure supplements for results in the crosswind direction, $y = z_2$. We train the algorithm by providing $N$ examples of input-output pairs $(\mathbf{x}_i, y_i)$ selected randomly from the full simulation, and obtain the function that connects input and output: $y \approx f(\mathbf{x})$.

We propose a machine learning approach where the different odor features are ranked based on their predictive power, rather than their fitting properties. Different data-sets of odor/distance pairs are defined. The data-sets differ in the way odor measurements are represented in terms of feature vectors. For each data-set we learn a function to predict the distance to target given the corresponding odor features. The predictive power of each function, and corresponding set of features, is

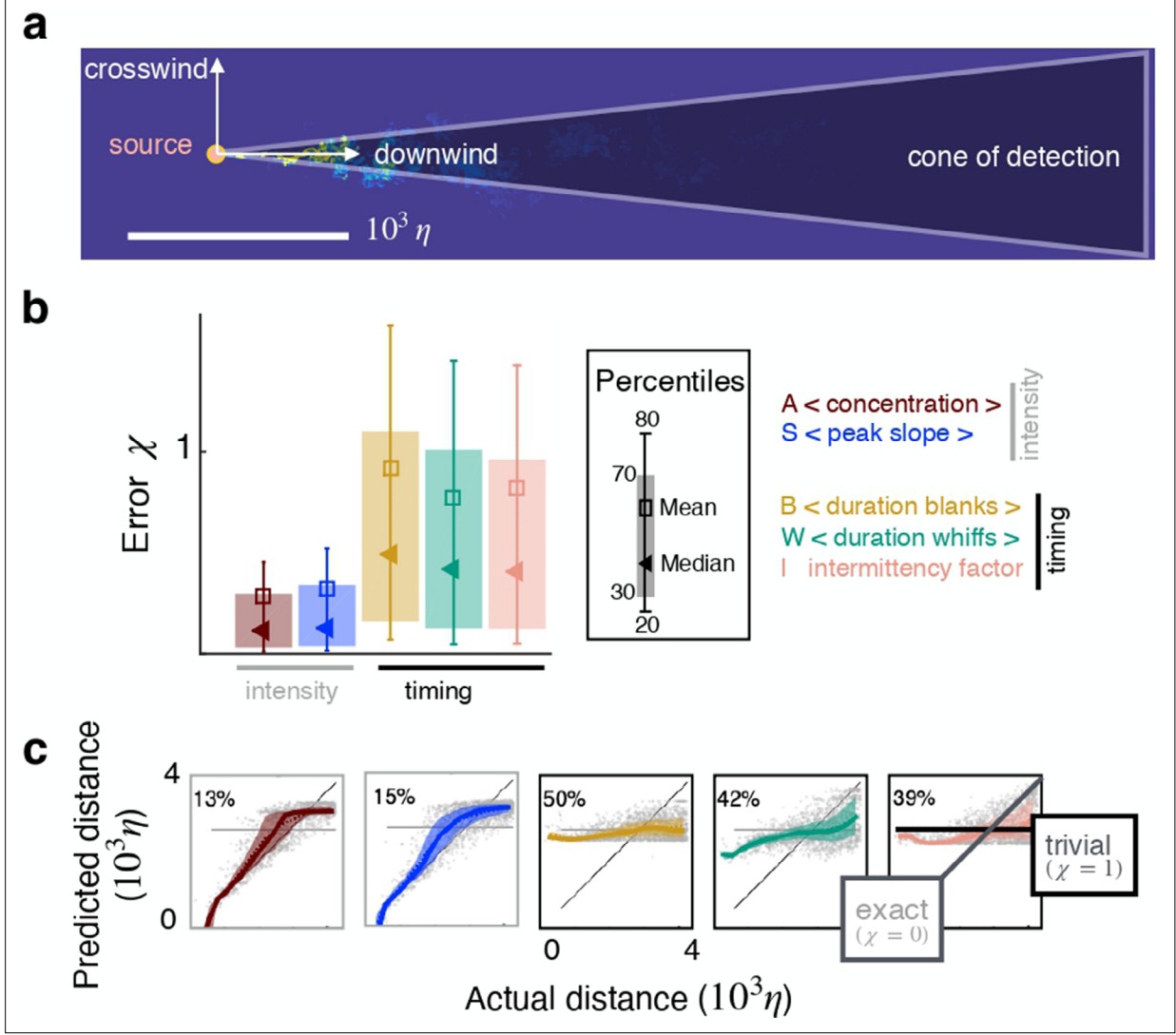

**Figure 2.** Individual features enable inference in two dimensions. (**a**) Sketch of the geometry. (**b**) Test error $\chi$ for inference using individual features as input. (**c**) Predicted vs actual distance for inference. Prediction for representative test points (grey circles); 30–70th percentile (patch, same color code as in (**b**)); trivial prediction $f(\mathbf{x}) = <y>_{\text{test}}$ (solid horizontal line, corresponds to $\chi = 1$); exact prediction (bisector, corresponds to $\chi = 0$); dispersion away from the bisector visualizes the prediction error. Results are obtained with a supervised learning algorithm based on regularized empirical risk minimization (Materials and methods). Each input datum $x_i$ is one individual scalar feature computed from the time course of odor concentration measured at location $\mathbf{z}_i$ at 100 evenly spaced time points with sampling frequency $\omega = 1/\tau_\eta$, where $\tau_\eta$ is Kolmogorov time. The training/test set are composed of $N = 5000$ and $N_t = 13500$ data points, respectively.

The online version of this article includes the following figure supplement(s) for figure 2:

**Figure supplement 1.** Fixed vs adaptive threshold.

**Figure supplement 2.** Linear least square algorithm.

**Figure supplement 3.** Visualisation of a typical cross validation procedure.

**Figure supplement 4.** Test error at source height, for prediction in the crosswind direction.

**Figure supplement 5.** Results for three source locations.

**Figure supplement 6.** Results of training with odor emanating from one source and testing over odor fields emanating from the other two.

then assessed. More precisely, each data-set is split in a training and a test set, as custom in machine learning. Training sets are used to learn functions connecting odor to target location, whereas test sets are used to assess their prediction properties. The training/test split is crucial since the goal is to make good predictions on new, unseen points, that are not within the training sets. From a modeling perspective, a flexible nonlinear/nonparametric approach based on kernel methods is contrasted and

shown to be superior to a simpler linear model (*Figure 2—figure supplement 2*). A careful protocol based on hold-out cross-validation is used to select the hyper-parameters of the considered learning models (we refer to Materials and methods and *Figure 2—figure supplement 3* for more details).

To illustrate the results we pick the two-dimensional plane at height $H/4$ that contains the source. The first result is that individual features ($d = 1$) bear useful information for two-dimensional source localization even at several meters from the source. Performance is quantified by the normalized squared error averaged over the $N_t$ points in the test set $\chi = \sum_{i=1}^{N_t} [y_i - f(\mathbf{x}_i)]^2 / \sum_{i=1}^{N_t} [y_i - \bar{y}]^2$. For this dataset, intensity features rank higher than timing features (*Figure 2b–c*), consistent with previous work *Atema, 1996* and predictions are more accurate in the crosswind than in the downwind direction (compare with *Figure 2—figure supplement 4*). For reference, a random guess with flat probability within the correct lower and upper bounds yields $\chi_{\text{random}} = 2$, whereas a target function $f_{\text{trivial}}(x) = \langle y \rangle_{\text{test}}$ that learns the average of the output over the test set yields $\chi_{\text{trivial}} = 1$.

In order to prove that the algorithm captures the dynamics of the scalar and not of the underlying velocity field, we realize three computational fluid dynamics simulations with the odor source at different locations downstream of the obstacle. Each source is placed at a different distance from the obstacle and thus feels different velocity fields, because the flow is inhomogeneous in the downstream direction. To perform a fair comparison we train the algorithm on points sampled from a conical domain based on the most downstream source; for the other two sources, we use an identical domain shifted upstream so that the vertex of the cone is located at the respective source location (see *Figure 2—figure supplement 5* left). Performance at source height varies little over the three locations, demonstrating that the algorithm is learning the dynamics of the scalar and not of the carrying flow (see *Figure 2—figure supplement 5* center). Additionally we demonstrate that, irrespective of source position, timing features acquire predictive power with height, whereas the opposite is observed for intensity features (see *Figure 2—figure supplement 5* right). We discuss this property in further detail in the following. . Finally, we train the algorithm with the odor fields emanating from one source location, and test its performance over odor fields generated by the two other sources. *Figure 2—figure supplement 6* shows that performance of individual features varies little when test and training are performed over different dataset. Learning from pairs of features appears somewhat more sensitive to the details of the dataset. In the aggregate, the analysis corroborates that the algorithm captures odor dynamics and is rather insensitive to details of the underlying flow. In the rest of the manuscript we focus on the leftmost source, so as to exploit the full spatial range available from the simulations.

Next we analyze whether and how the sampling strategy affects performance and ranking of the features. Most results are shown for a memory of $100\tau_\eta \approx 15\,s$. Performance improves with longer memory (*Figure 3a*), because this allows to better average out noise and obtain more stable estimates of the features. But improvement follows a slow power law so that waiting for example 20 times longer yields predictions only about twice as precise. On the other hand, waiting as little as $10\tau_\eta \approx 1.5$ seconds still allows to make predictions, albeit less precise. We then verify whether performance may improve with a larger training set. Because we infer distance from an individual (scalar) feature, the problem is one dimensional and we find that a small number of training points, which we indicate with $N$, is sufficient to reach a plateau in prediction performance (*Figure 3b*). We choose $N = 5000$ training points, which is also robust to the case with more than one feature (*Figure 3—figure supplement 1*). Finally, sampling more frequently than once per Kolmogorov time does not essentially affect the results nor ranking (*Figure 3c*). Similar results hold for the crosswind direction (*Figure 3—figure supplement 2*).

Pairing two observables improves performance in some cases, but not always. In fact, pairing two features of the same category results in little to no improvement (*Figure 4* and similarly for the crosswind direction, *Figure 4—figure supplement 1*). In contrast, combining one intensity and one timing feature improves performance considerably, up to 65%. This result can be understood by mapping the error done by individual features in space (*Figure 5—figure supplement 1*), showing that intensity and timing features are complementary, that is intensity features perform well in locations where timing features perform poorly.

We next seek to clarify whether the results depend on space. To this end we compose five different dataset, *a* to *e*, obtained by extracting odor snapshots from horizontal planes at source height (*b*), above the source (*c* to *e*), and below the source (*a*) (*Figure 5a*). From *a* to *e*, sparsity increases and

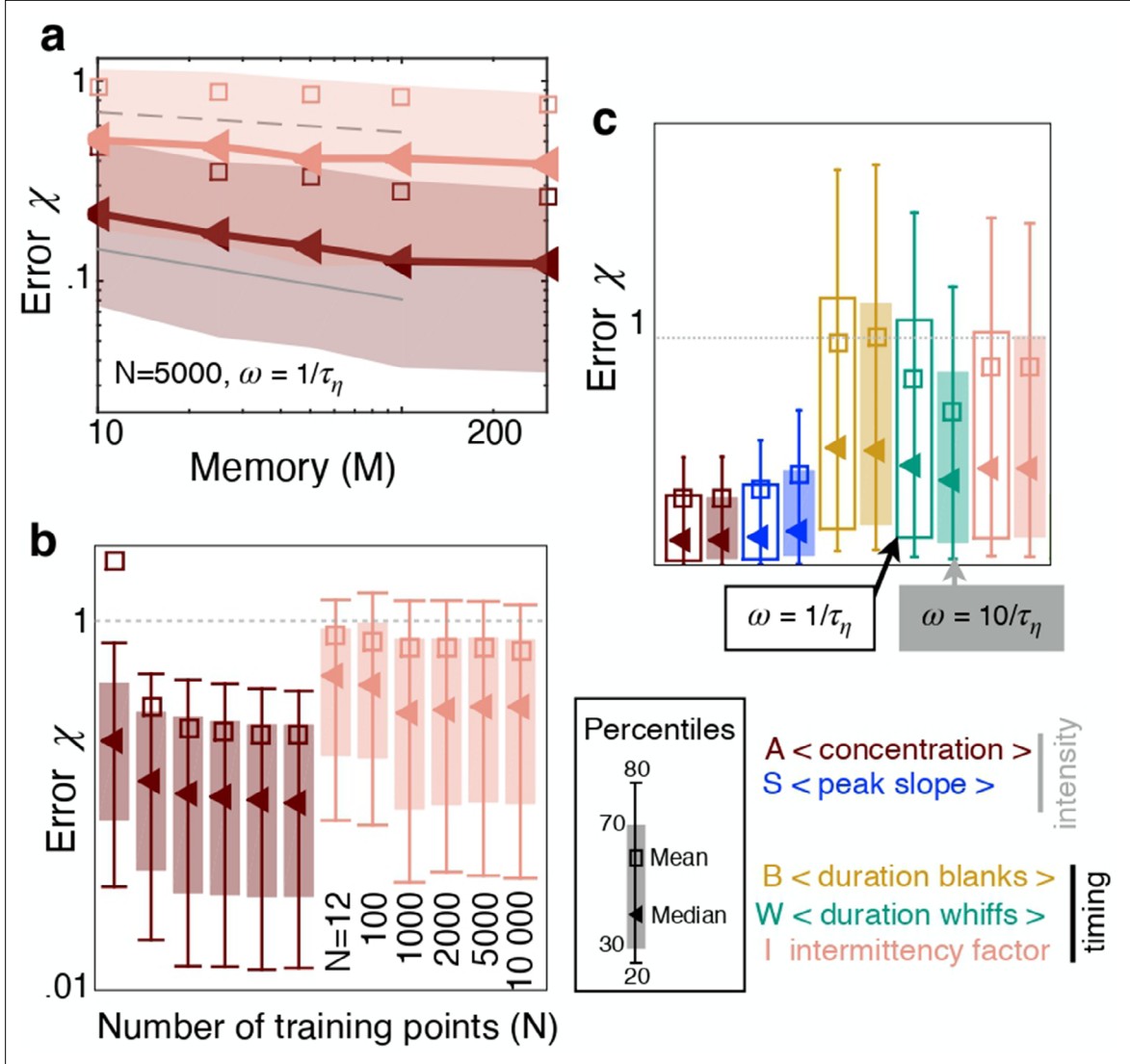

**Figure 3.** The sampling strategy affects performance but not ranking. (**a**) Error $\chi$ as a function of memory in units of Kolmogorov times $\tau_\eta$; memory is defined as the duration of the time series of odor concentration $\mathbf{c}_i = (c(\mathbf{z}_i, t_i), ..., c(\mathbf{z}_i, t_{i+M}))$ used to compute the five features $x_i^1, ..., x_i^5$, i.e. memory $= t_{i+M} - t_i = M/\omega$. Red and pink: Performance using $\mathbf{x}_i = x_i^1$ (average concentration) and $\mathbf{x}_i = x_i^5$ (intermittency factor). The number of training points and the frequency of sampling are fixed, $N = 5000$ and $\omega = 1/\tau_\eta$. Dotted, dashed and solid grey lines are power laws with exponents $-1/5$, $-1/10$ and $-1/4$ respectively to guide the eye. (**b**) Error as a function of number of points in the training set $N$, with $N_t = 13500$ points in the test set, memory $= 100\tau_\eta$ and $\omega = 1/\tau_\eta$. Color code as in (**a**). (**c**) Performance using the five individual features as input with $N = 5000$, $N_t = 13500$, memory $= 100\,\tau_\eta$ sampling odor at frequency $\omega = 1/\tau_\eta$ (empty bars) and $\omega = 10/\tau_\eta$ (filled bars). Key shows color coding.

The online version of this article includes the following figure supplement(s) for figure 3:

**Figure supplement 1.** Prediction error as a function of the number of points in the training set for individual features and pairs of features.

**Figure supplement 2.** Effect of memory, sampling frequency and number of points in the training set, for prediction in the crosswind direction.

intensity decreases (*Figure 5b*) simply because closer to the boundary, the air slows down and the odor accumulates. By analyzing performance across these dataset, we find that ranking of individual features shifts considerably. The two intensity features outperform all timing features when the dataset is not very sparse (dataset *a-b*, *Figure 5c and d* left). In contrast, two timing features (intermittency factor and blank duration) outperform all others for the more sparse and less intense dataset *d-e* (*Figure 5c and d* right). Whiff duration performs poorly in *d-e* because intermittency is too severe and whiffs are short in duration thus bear little information (the average whiff duration is 1–7 time steps in over 90% of the time series). Although the ranking of individual features shifts with height, pairing one intensity and one timing feature remains the most successful strategy across all heights

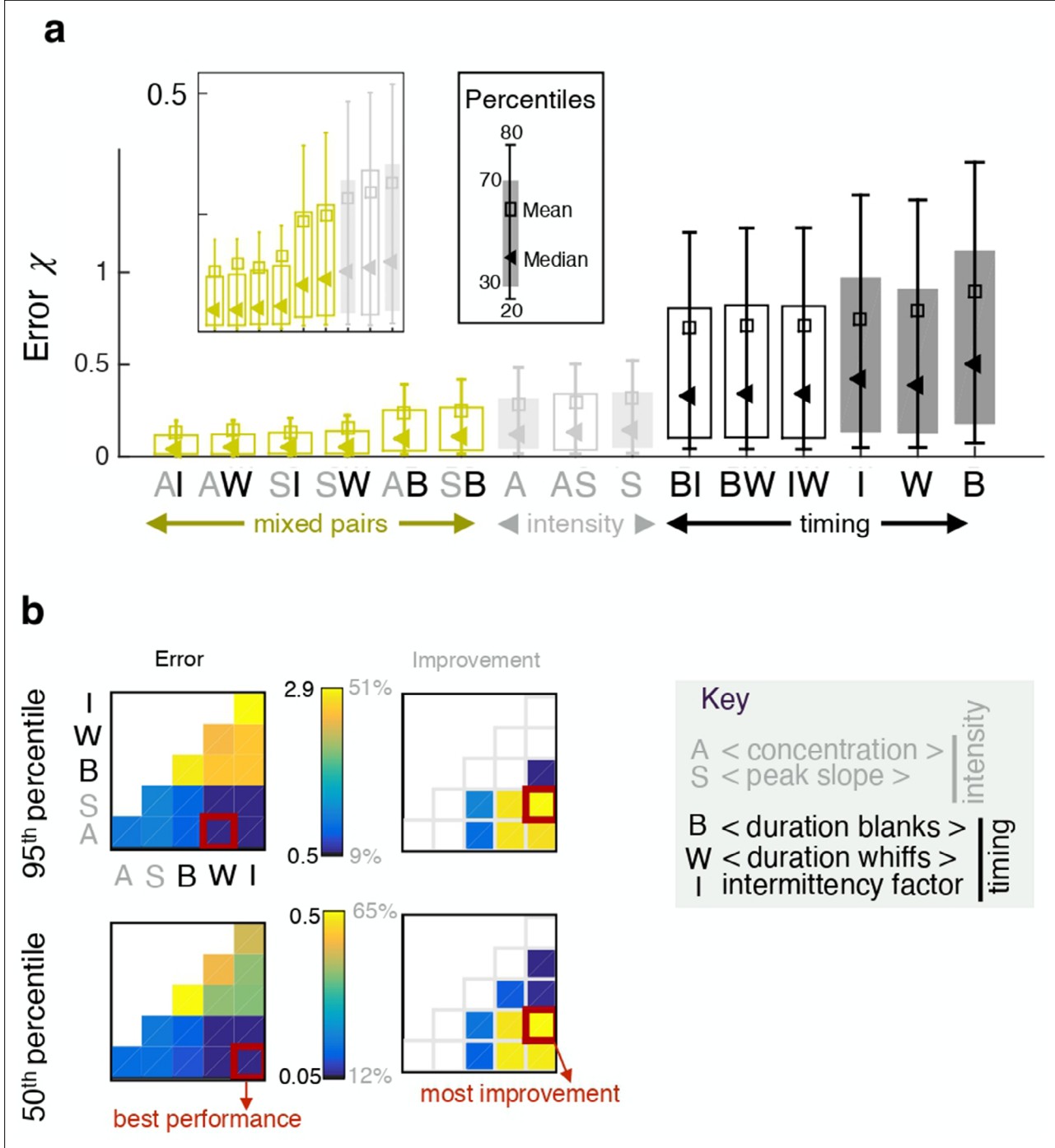

**Figure 4.** Pairing one timing feature and one intensity feature considerably improves performance. (**a**) Error $\chi$ obtained with individual features (full bars) and pairs of features (empty bars). Grey and black indicate pairings of two intensity features and two timing features respectively; green indicates mixed pairs of one timing and one intensity feature. (**b**) Performance (left) and relative improvement over the best of the two paired features (right). Results for the median (bottom) and the 95th percentile (top). Within each table plot, rows from bottom to top and columns from left to right are labeled by the 5 individual features: A (average, $x^1$), S (slope $x^2$), B (blanks $x^3$), W (whiffs $x^4$), I (intermittency $x^5$). Results with individual features are shown on the diagonal; results pairing feature and feature $j$ are shown at position $(i, j)$. Mixed pairs provide both the best performance and the largest improvement over individual features.

The online version of this article includes the following figure supplement(s) for figure 4:

**Figure supplement 1.** Effect of pairing two individual features, for prediction in the crosswind direction.

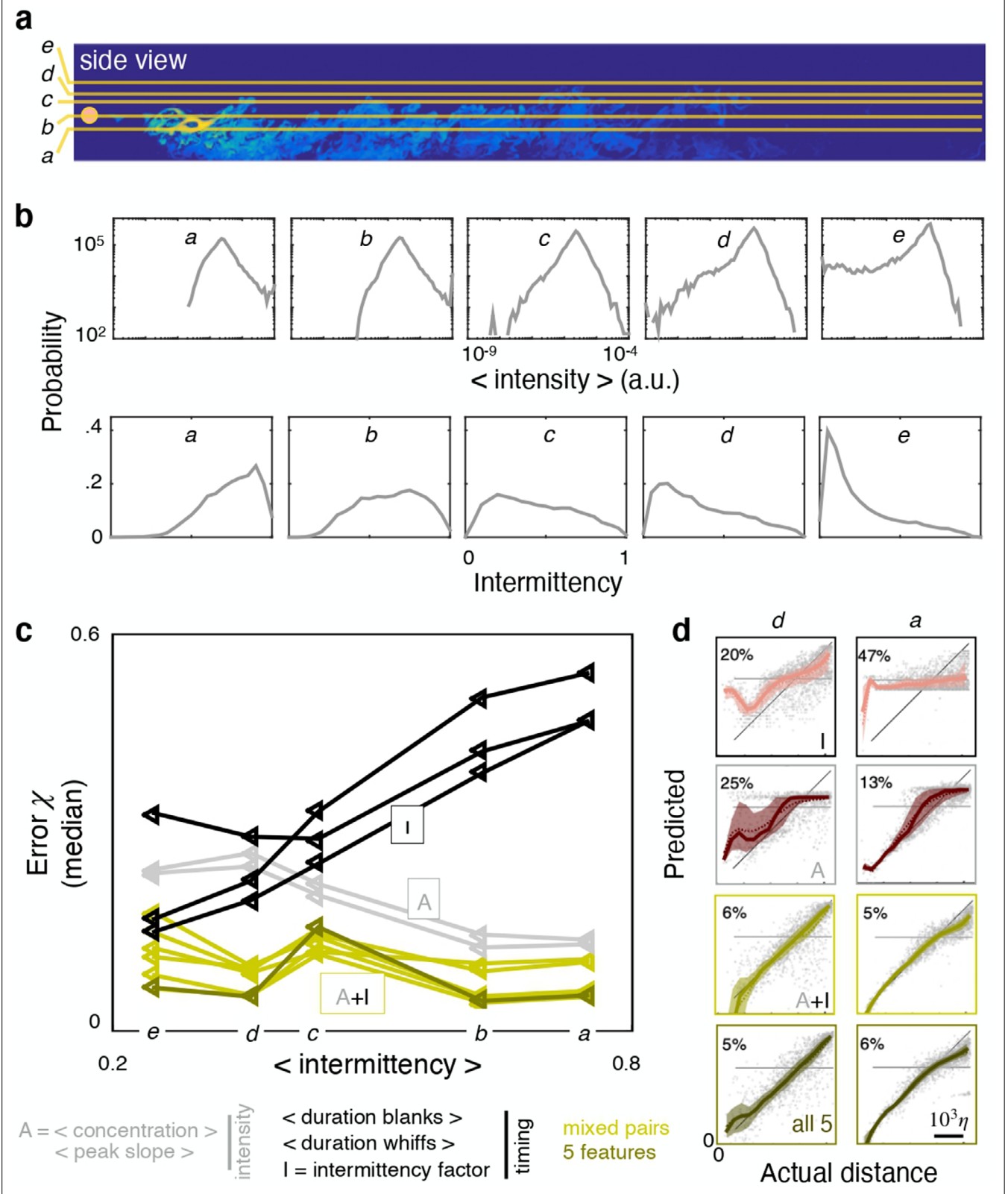

**Figure 5.** Ranking shifts with height from the ground. (**a**) Datasets *a* to *e* correspond to data obtained at heights *z/H* = 25%, 37.5%, 50%, 55% and 65% respectively. (**b**) Distribution of intensities (top) and intermittency factors (bottom) over the training set from *a* to *e* (left to right). Moving away from the boundary, the odor becomes less intense and more sparse. (**c**) Median performance as a function of average intermittency factor of the training set for individual intensity (grey) and timing (black) features, mixed pairs of one intensity and one timing feature (green) and all five features together

*Figure 5 continued*

(dark green). (**d**) Predicted *vs* actual distance, to visualize a representative subset of the results in (**c**), scale bar $10^3\eta$. Ranking depends sensibly on height: intensity features outperform timing features near the substrate, where there is more odor and it is more continuous; timing features outperform intensity features further from the substrate where there is less odor and it is more sparse; mixed pairs perform best across all conditions; combining five features provides little to no improvement over mixed pairs.

The online version of this article includes the following figure supplement(s) for figure 5:

**Figure supplement 1.** Test error mapped in space.

**Figure supplement 2.** Predicted distance vs actual distance for simulation a (top) and (**d**) (bottom); symbols as in *Figure 5d*.

**Figure supplement 3.** Estimated predicted power of individual features.

(*Figure 5c and d*). In contrast, combining all five features contributes little improvement (*Figure 5— figure supplement 2*).

Let us now focus on the plane at source height and separate locations based on their distance from the source. We assemble a distal dataset and a proximal dataset, composed of points that are further and closer than 2330 $\eta$ from the source respectively (*Figure 6a*). The odor is more intense and more sparse closer to the source and it becomes more dilute and less sparse with distance from the source (*Figure 6b*). Performance of individual features degrades with distance (*Figure 6d*). Intensity features clearly outperform timing features at close range, as seen both from various percentiles of the test error (*Figure 6d*, left) as well as the full distribution (*Figure 6c*, left). The disparity between timing and intensity features disappears in the distal problem: the error distribution for all individual features is essentially superimposed except for the tails (*Figure 6c*, right and inset), which cause small differences in the median and other percentiles of the error (*Figure 6d*, right). Remarkably, mixed pairs outperform all individual features in both the distal and proximal problems (*Figure 6c–d*). In the aggregate, results demonstrate that, even within a single turbulent flow, ranking shifts considerably. Namely, measuring timing of odor encounters is most useful in regions where the odor is dilute, that is far from the source and from the substrate, whereas measuring intensity is most useful in concentrated conditions, that is close to the source or the substrate.

Thus, the predictive power of different features varies greatly in space and time. Next, we show how this spatial variation is dictated by the statistical properties of the odor plume. To this end, we provide an analytical characterization of the test error of individual features, $\chi$, that connects directly to the physics of the problem. Such a characterization is consistent with the observed variation in performance. Note that for large enough samples, the test error $\chi$ approximates the following expected error

$$\frac{\int_0^\infty dx \int_0^R dy (y - f^*(x))^2 p(x,y)}{\int_0^R (y - \bar{y})^2 p(y) dy}. \tag{1}$$

The latter is called expected error and is the ideal version of the test error. It can be interpreted as the error summed over all possible input-output pairs, weighted by their corresponding joint probability to be sampled. Here, $f^*(x) = \langle y|x \rangle = \int_0^R yp(y|x)dy$ is the so called regression function, which minimizes the expected error among all possible functions. The regression function can be approximated using Kernel ridge regression and sufficiently rich kernels. Indeed, kernel ridge regression is known to be a so called universal estimator *Hastie et al., 2001*; *Steinwart, 2002*. In the above expression, $R$ is the length of the cone, $p(y|x)$ is the conditional probability distribution of the output given the input, $p(x,y)$ is the joint probability distribution of the input output pair $(x,y)$ is the prior distribution on the output $y$. The idea is to relate $f^*$, hence the expected error, to the distribution $p(x|y)$ of observing feature $x$ depending on distance $y$. Indeed, the latter is dictated by the fluid dynamics of odor plumes from a concentrated source, and hence provides a more direct connection between the expected error and the physics of the problem. Since the considered features are sample averages, in the limit of large samples, their distribution is well approximated by a Gaussian, hence fully characterized by the mean and standard deviation. Then simple estimates can be computed empirically from our data. *Equation (1)*, provided with these estimates, reproduces the predictive power of individual features showed in *Figure 5c* (see *Figure 5—figure supplement 3*). Note that whiffs deviate considerably from a normal distribution, hence the argument needs to be revised for this feature. To move beyond

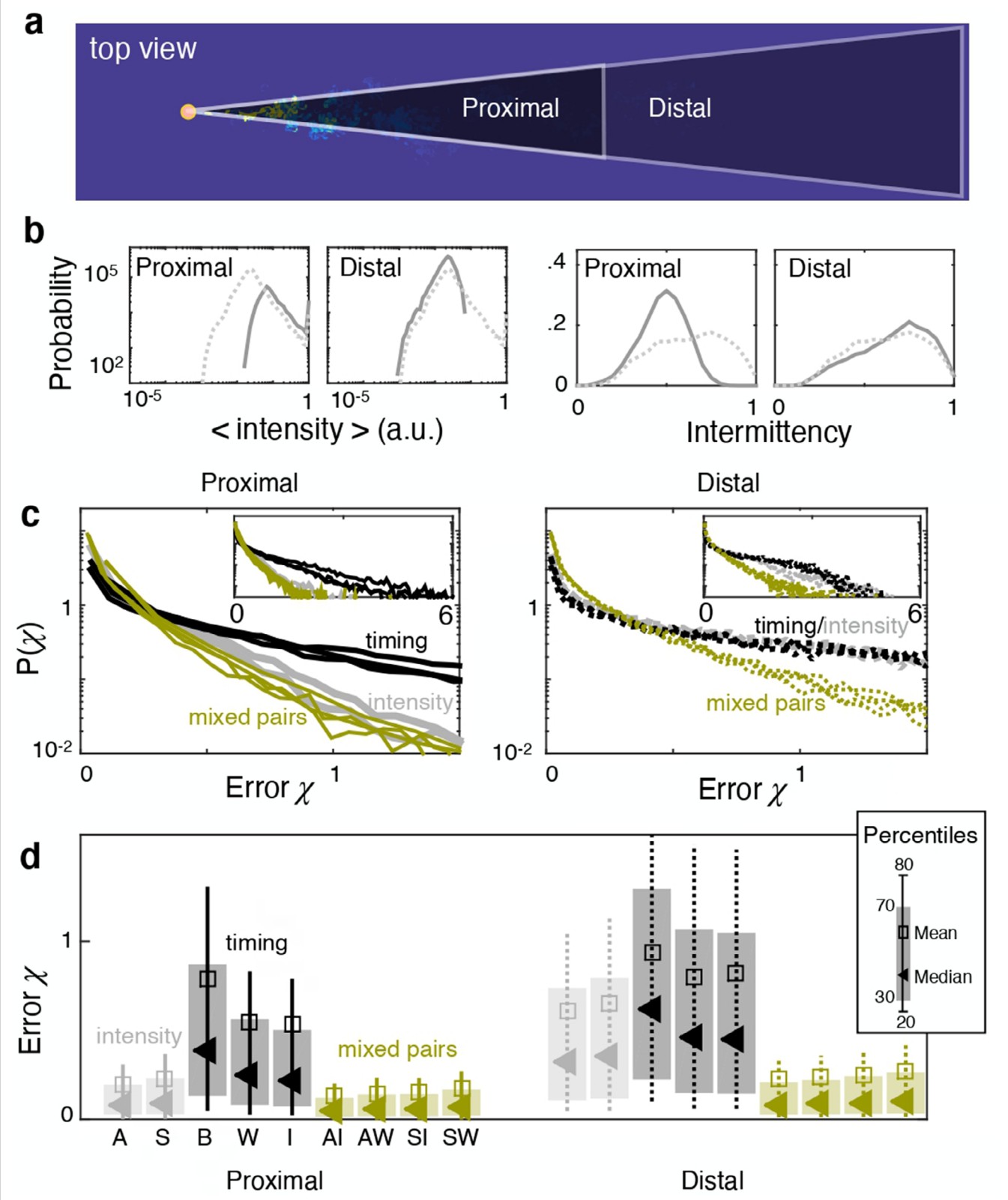

**Figure 6.** Ranking depends on distance from the source. (**a**) At source height, the dataset is split in proximal (distance <2330 $\eta$ ) and distal (distance >2330 $\eta$ ). (**b**) Distributions of average odor intensity (left) and intermittency factor (right) over the training set; closer to the source, the odor is more intense and more sparse. (**c**) Distribution of test error for the proximal (left) and distal (right) problem showing intensity features (grey) outperform timing features (black) at close range, but not in the distal problem where differences in the error distribution are limited to the tails (see insets). Mixed

*Figure 6 continued on next page*

*Figure 6 continued*

pairs of features (green) outperform individual features either marginally (left) or considerably (right). (**d**) Percentiles of the error distribution in (**c**) for the proximal (left) and distal (right) problems confirming the picture emerged from (**c**).

## Discussion

Our results demonstrate that within the cone of detection, the time course of an odor bears useful information for source localization even at meters from the source. We find that the concentration and the slope of a turbulent odor signal, averaged over a memory lag, are particularly useful to predict source location at close range or near the boundary. These features quantify the intensity of the odor and its variation. The primacy of the intensity features wanes in more challenging conditions, for example moving away from the source or away from the boundary. In these portions of space, where the odor is scarcer, features that quantify timing of odor detection become as effective as intensity features, or more effective. One of the best studied example of olfactory search in dilute conditions is arguably the case of insects.

Interestingly, olfactory receptor neurons in insects appear to encode efficiently information about timing across a wide range of intensities *Gorur-Shandilya et al., 2017*; *Martelli et al., 2013*.

Note that while the statistics of an odor plume clearly depends on all details of the flow and the source, see e.g. *Justus et al., 2002*; *Celani et al., 2014*; *Fackrell and Robins, 1982*, here we keep all of these parameters constant and demonstrate that even within a single flow, odor dynamics and the best predictors vary considerably in space. This begs the next question: do organisms switch between different modalities depending on attributes of odor dynamics, which will vary in space? This could be the case for mice, where the neural activity in the first relay of olfactory processing does in fact depend on how sparse is the odor *Lewis et al., 2021*. Specifically, sparse odor cues elicit individual responses that follow closely the ups and downs of the odor in time. In contrast, continuous signals elicit intense responses which are however uncorrelated to the temporal dynamics of the odor itself *Lewis et al., 2021*.

We find that features within the same class are redundant whereas features from different classes are complementary. Indeed, features of the same class have similar patterns of performance in space, but each class has a distinct pattern. As a consequence, measuring both timing and intensity is beneficial, but using more than one feature to quantify either timing or intensity provides no advantage. Combining all features does not improve over the performance of mixed pairs, consistent with redundancy within each class. Note that there is no fundamental reason to expect features from the same class to be redundant, and further work with a larger library of features is needed to prove or disprove this notion.

Importantly, mixed time/intensity pairs of features outrank individual features robustly, that is in all portions of space, regardless of distance from the source and from the ground. This is in contrast with individual features and suggests relying on simultaneous timing and intensity features is advantageous when odors are sensed at various distances from the source and from the substrate. Interestingly, the coexistence of bursting olfactory neurons and canonical olfactory neurons in lobsters suggests these animals are in fact able to measure simultaneously timing and intensity *Park et al., 2014*; *Ache et al., 2016*, which is consistent with the increased predictive power of the mixed pairs of features. Similarly, in mammals, optogenetic activation of the olfactory bulb *Smear et al., 2013* demonstrates that both kinds of measures guide behavior (lick *vs* no lick).

In this work, we have investigated the problem of predicting the location of a target from measures of the time course of a turbulent odor. Previous work explored a related question, that is how to best represent instantaneous snapshots of the odor to encode maximum information about source location *Victor et al., 2019*. The two approaches are not immediately comparable: first, *Victor et al., 2019* consider few snapshots of the odor, rather than measures of its time course. Second, maximizing information does not guarantee good predictions (to make predictions information needs to be extracted and processed, and importantly the focus is on new data that were not previously seen). We provide

two comments that are relevant if information is the limiting factor for prediction accuracy: (*i*) binary representations were suboptimal in all conditions considered in *Victor et al., 2019*; *Boie et al., 2018*, that is at few tens of cm from the source. This is consistent with our results in concentrated conditions, where timing features -accessible through binary representations- are suboptimal. Our evidences suggest, however, that the result may not hold in more dilute conditions, where the gap between binary and more accurate representations should become increasingly small. (*ii*) Individual snapshots of odor from *Victor et al., 2019*; *Boie et al., 2018* contained 1–2 bits of information about source location, but allocating more resources to represent how the odor varies in time was found informative *Victor et al., 2019*; *Boie et al., 2018*. Our mixed pairs of features at close range achieve precisions of 5–6%, corresponding to coding for position with words of 4–4.3 bits. Our results thus confirm that memory is indeed useful, but the gain does not increase indefinitely with further memory.

The literature on olfactory navigation is vast. Although a complete review of available algorithms is beyond the scope of the present work, we remark that recent results investigated gradient descent algorithms using either concentration alone *Gire et al., 2016*, or various measures of timing and intensity *Park et al., 2016*; *Michaelis et al., 2020*; *Leathers et al., 2020*. Overall, both intensity and timing appear to have a potential to lead to an odor source, consistent with our results on individual features. A combination of the two kinds of features was found beneficial in *Leathers et al., 2020*, consistent with our results on mixed pairs. Note that predicting source location and navigating to reach it are distinct tasks. Although they are often assumed to be intimately connected, whether good predictors may be good variables for navigation in more general contexts remains to be understood.

Here, we have analyzed the features that enable the most accurate prediction of source location. We add a few observations about the significance of the results for animal behavior. First: whether animals rely on features from either class will depend on what features best support behavior. It is often implicitly assumed that features that bear reliable information on source location are also the most useful for navigation. However, this connection between prediction and navigation is far from straightforward and more work is needed to establish whether accurate predictions imply efficient navigation. Second: animals are unlikely to have prior information on the details of the odor source, for example its intensity. Timing features are more robust than intensity features with respect to the intensity of the source and may thus be favored regardless of their performance, which was argued in *Schmuker et al., 2016*. In our work, timing features are precisely invariant with source intensity because we define the detection threshold adaptively (see Materials and methods). More realistic conditions will need to be evaluated, where dependence on source intensity emerges as a result of non-linearities that we did not model in this work. These effects emerge for example, close to a boundary which partially absorbs the odor *Gorur-Shandilya et al., 2019*, or in the case of fixed thresholds, although this dependence is weak in the far field where timing features are most useful *Celani et al., 2014*. Third: we have focused on predicting source location from within the cone of detection, where an agent will detect the odor quite often. However, a crucial difficulty of turbulent navigation is to find the cone itself. We cannot address the problem of predicting source location from outside the cone because detections are so rare that we lack statistics. The distinction between inside and outside the cone of detection is key for navigation with sparse cues (see *Reddy et al., 2021*) and deserves further attention.

## Materials and methods
### Direct numerical simulations of turbulent odor plumes

To reproduce a realistic odor landscape and generate the dataset showed in *Figure 1*, we solve the Navier-Stokes (2) and the advection-diffusion equation for passive odor transport (3) at all relevant scales of motion from the smallest turbulent eddies (Kolmogorov scale $\eta$) to the integral scale ($L > 600\eta$), using Direct numerical simulations (DNS):

$$\partial_t \mathbf{u} + \mathbf{u} \cdot \nabla \mathbf{u} = -\frac{1}{\rho}\nabla P + \nu \nabla^2 \mathbf{u} \qquad \nabla \cdot \mathbf{u} = 0 \tag{2}$$

$$\partial_t c + \mathbf{u} \cdot \nabla c = \kappa_c \nabla^2 c + q \tag{3}$$

where $\mathbf{u}$ is the velocity field, $\rho$ is the fluid density, $P$ is pressure, $\nu$ is the fluid kinematic viscosity, $c$ is the odor concentration, $\kappa_c$ is its diffusivity and $q$ an odor source. All parameters are listed in *Table 1*.

Note that we use $Sc = 1$ which is appropriate for typical odors in air but not in water. However, we expect a weak dependence on the Schmidt number as the Batchelor and Kolmogorov scales are below the size of the source and we are interested in the large scale statistics *Falkovich et al., 2001*; *Celani et al., 2014*; *Duplat et al., 2010*. We simulate a turbulent channel flow with a concentrated odor source and an obstacle that generates turbulence by customizing the open-source software Nek5000 *Fischer et al., 2008* developed at Argonne National Laboratory, Illinois. Nek5000 employs a spectral element method (SEM) *Patera, 1984 Orszag, 1980* based on Legendre polynomials for discretization *Ho, 1989*, and a 4th order Runge-Kutta scheme for time marching. The code is written in fortran77 and C and it uses MPI for parallelization.

The three-dimensional channel is divided in $E = 160\,000$ discrete elements: $200 \times 40 \times 20$ (number of elements in length × width × height); within each element the solution is expanded in 8th grade tensor-product polynomials so that the domain is effectively discretized in 81 920 000 elements. The average spatial resolution is equal in each direction $\Delta x \approx 4\eta$. A cylindrical cap of height $= 160\,\eta$ is added on the ground; the cylinder spans the entire width of the channel. The mesh is adapted to fit the cylinder. Fluid flows from left to right and the obstacle generates turbulence in the channel, in particular the height of the cylinder tunes the velocity fluctuations. The velocity fluctuations are defined as $\delta u(\mathbf{z}, t) = u(\mathbf{z}, t) - \langle u(\mathbf{z}, t) \rangle_y$; their intensity is $u' = \sqrt{\langle (\delta u)^2 \rangle}$, where averages are intended in space and time. *Table 1* summarizes the parameters that characterize turbulence.

Each simulation runs for 300,000 time steps where $\delta t = 10^{-2}\tau_\eta$ and follows from a severe Courant criterium with $U\Delta t/\Delta x < 0.4$ to ensure convergence of both the velocity and scalar fields. Snapshots of velocity and odor fields are saved at constant frequency $\omega = 1/\tau_\eta$ (except for results in *Figure 3c* where snapshots are saved 10 times more frequently). Each DNS requires 2 weeks of computational time using 320 cpus.

## Boundary conditions and odor source

We impose a Poiseuille velocity profile at the inlet: $\mathbf{u} = (u, 0, 0)$ and $u = 6U_b(\zeta - \zeta^2)$, where $\zeta = z_3/H$ is the vertical coordinate normalized to the height of the channel and $U_b$ is the mean speed. We set a no-slip condition $\mathbf{u} = 0$ at the ground and on the obstacle; on the remaining boundaries we impose the turbulent outflow condition defined in *Fischer et al., 2007* that imposes a positive exit velocity to avoid potential negative flux and the consequent instability it generates.

More precisely, the divergence ramps up from zero to a positive value along the element closest to the boundary: $\nabla \cdot \mathbf{u} = C[1 - (z_\perp/\Delta x)^2]$, where $z_\perp$ is the distance from the boundary and $C = 2$ is the minimal value that ensures convergence. For the odor, we impose a Dirichlet condition ($c = 0$) at the ground, on the obstacle and at the inlet; while an outflow condition is set at the top, on the sides and at the outlet: $k(\nabla c) \cdot \mathbf{n} = 0$. We introduce a source located right above and downstream of the obstacle, at coordinates $x_s = 810\,\eta$, $y_s = 650\,\eta$, $z_s = 238\,\eta$; odor intensity at the source is defined by a gaussian distribution $q = e^{[(z_1 - x_s)^2 + (z_2 - y_s)^2 + (z_3 - z_s)^2]/(2\sigma^2)}$, where $\sigma = 5\eta$.

## Machine learning

To learn the correct position of a target source given an odor, we propose to use supervised machine learning. We next review some key ideas, and refer to standard textbooks for further details for example *Hastie et al., 2001*.

The goal in supervised learning is to infer a function $f$ given a training set $(\mathbf{x}_1, y_1), \ldots (\mathbf{x}_N, y_N)$ of input/output pairs. A good function estimate should allow to *predict* the outputs associated to *new* input points. In our setting each input $\mathbf{x}$ is a one-, two-, or five-dimensional vector whose entries are scalar features of odor time series, where the odor is sampled at a specific spatial location. From every sampling location, we compute the distance to the source and this distance is the output $y$.

To measure how close the prediction $f(\mathbf{x})$ is to the correct output $y$, we consider the square loss $(f(\mathbf{x}) - y)^2$. Following a statistical learning framework, the data are assumed to be sampled according to a fixed but unknown data distribution $P$. In this view, the ideal solution $f^*$ should minimize the expected loss $\langle l(f(\mathbf{x}), y) \rangle$ over all data distributed according to $P$. In practice, only an empirical loss based on training data can be measured, and the search for a solution needs be restricted to a suitable class of hypothesis. Note that, the choice of the latter is critical since the nature of the function to be learnt is not known a priori. A basic choice is considering linear functions $f(\mathbf{x}) = \mathbf{w} \cdot \mathbf{x}$. In this case, minimizing the training loss reduces to linear least squares $\min \frac{1}{N}\|Y - X \cdot \mathbf{w}\|^2$, where $X$ is the matrix

composed of the $N$ training data input $X = (\mathbf{x}_1, ..., \mathbf{x}_N)^T$ and $Y$ is the vector composed of the $N$ labels of the training set $Y = (y_1, ..., y_N)^T$. The corresponding solution is easily shown to be $\mathbf{w} = (X^T X)^{-1} X^T Y$. In *Figure 2—figure supplement 1*, we show that the choice of linear models has limited predictive power and does not allow to rank features. To tackle this issue, we consider kernel methods *Schölkopf and Smola, 2002*, a more powerful class of nonlinear models corresponding to functions of the form $f(\mathbf{x}) = \sum_{i=1}^{N} k(\mathbf{x}_i, \mathbf{x}) \mathbf{c}_i$. Here, $k(\mathbf{x}, \mathbf{x}')$ is a so called kernel, that here we will choose to be the Gaussian kernel $k(\mathbf{x}, \mathbf{x}') = e^{-\|\mathbf{x}-\mathbf{x}'\|^2/2\sigma^2}$. The coefficients $\mathbf{c} = (\mathbf{c}_1, \ldots, \mathbf{c}_n)$ are given by the expression

$$\mathbf{c} = (K + \lambda N I)^{-1} \mathbf{y} \tag{4}$$

which minimizes

$$\frac{1}{N} \|K\mathbf{c} - \mathbf{y}\| + \lambda \mathbf{c}^\top K \mathbf{c}.$$

In the above expression, $K$ is the $N$ by $N$ matrix with entries $K_{ij} = k(\mathbf{x}_i, \mathbf{x}_j)$. The first term can be shown to be a data fit term whereas the second term can be shown to control the regularity of the obtained solution *Schölkopf and Smola, 2002*. The *regularization parameter* $\lambda$ balances out the two terms and needs be tuned, together with the kernel parameters (the Gaussian width $\sigma$ in our case).

Kernel methods offer a number of advantages. They are nonlinear, and hence can learn a wide range of complex input/output behavior. They are an example of nonparametric models, where the complexity of the model can adapt to the problem at hand and indeed learn any kind of continuous function provided enough data. This can be contrasted to linear models that clearly cannot learn any nonlinear function. Moreover, by tuning the hyper-parameters $\lambda, \sigma$ more or less complex shape can be selected. When $\lambda$ is small we are simply fitting the data, possibly at the price of stability, whereas for large $\lambda$ we are favoring simpler models. With small $\sigma$ we allow highly varying functions, whereas with large enough $\sigma$ we essentially recover linear models.

Indeed, the choice of these parameters is crucial and tested and visualized in *Figure 2—figure supplement 3*. Here it is shown that for $\lambda \to 0$, the solution incurs in the well known stability issues for large $\sigma$ and overfitting issues for small $\sigma$. We note that ideally one would want to choose these hyper-parameters minimizing the test error; however, this would lead to overoptimistic estimates of the prediction properties of the obtained model. Hence, we consider a hold-out cross validation protocol, where the training data are further split in a training and a validation sets. The new training set is used to compute solutions corresponding to different hyper-parameters. The validation set is used as a proxy for the text error to select the hyper-parameters with small corresponding error. The prediction properties of the model thus tuned is then assessed on the test set.

## Dataset

To compose the dataset for regression we first extract two-dimensional snapshots of odor at fixed height from the 3D simulation. Each snapshot from the simulation has dimensions $1600 \times 320$ (number of points in the downwind direction × crosswind direction). The initial evolution up to $300 \, \tau_\eta$ is excluded from the analysis as odor has not yet reached a stationary state. At stationary state we save 2700 frames at frequency $\omega = 1/\tau_\eta$ per simulation. Thus at each spatial location we have the entire time evolution composed of 2700 time points at regular intervals of $\tau_\eta$. We partition each simulation in fragments with $M$ snapshots (duration $M\tau_\eta$). Most simulations are shown for $M = 100$, thus for each spatial location we have 27 time series of the same duration (except for results leading to *Figure 3a*, where we vary memory from $10\tau_\eta$ to $250\tau_\eta$ resulting in 270 to 10 time series per location respectively).

The characteristic shape of the odor plume is a cone (*Figure 2*), that we defined as the region where the probability of detection computed over the entire simulation is larger than 0.35. The training set and test set are obtained by extracting $N = 5000$ (unless otherwise stated) and $N_t = 13500$ time series portions of duration $M\tau_\eta$. To select these $M$-long time series, we extract random locations $\mathbf{z}_i$ to cover homogeneously the cone, i.e. with flat probability within the cone, and random initial times $t_i$, with the training in the first half of the time history and the test in the second half of the time history. Time series that remain entirely under threshold are excluded.

Each odor time series is further processed by computing five features, two of which quantify intensity of the odor and rely on a precise representation of odor concentration (average concentration and average peak slope) and three of which quantify timing of odor encounters and are computed

after binarizing the odor (average whiff and blank duration and intermittency factor). The threshold $c_{thr}$ used for binarization is adaptive i.e. $c_{thr} = 0.5\langle c|c > 0\rangle$, where the average is computed over each time series separately. The threshold thus varies from $c_{thr} = 0.5c_0$ at the source to $c_{thr} = 10^{-6}c_0$ at the farthest edges of the cone, where $c_0$ is the concentration at the source. The choice of an adaptive threshold was suggested in *Gorur-Shandilya et al., 2017*. The precise value of the relative threshold has little effect on the results as shown in *Figure 2—figure supplement 1*, left. Fixed thresholds were tested and discarded because results depend sensibly on the threshold and the optimal threshold varies with the dataset in non-trivial ways (*Figure 2—figure supplement 1*, right). Finally, adaptive thresholds that are defined based on purely local information appear more plausible for a biological system that has no information on the intensity of the source.

The parameters $\lambda$ and $\sigma$ are obtained through 4-folds cross validation: the training set is split in 4 equal parts, 3 are used for training and 1 for validation. The empirical risk is computed on the validation set and averaged over the 4 possible permutations, systematically varying the hyperparameters $\lambda, \sigma$. The couple of hyperparameters that minimize the empirical risk over the validation set is selected through grid search using an $8 \times 8$ regular grid and further refined with a $4 \times 4$ subgrid. Results are insensitive to further refinement because there is a large plateau around the minimum, as shown in *Figure 5—figure supplement 2*. The optimal hyperparameters are used to compute the solution (4). The error $\chi$ used throughout the manuscript is simply the normalized test error $\chi = \sum_{i=1}^{N_t}[y_i - f(\mathbf{x}_i)]^2 / \sum_{i=1}^{N_t}[y_i - \bar{y}]^2$. For most of the figures, we used 5000 training points and up to 13,500 testing points (we removed from the dataset all the points with null entries for the entire time span), we implemented Kernel ridge regression using FALKON *Rudi et al., 2018*, a fast algorithm for matrix inversion (the number of iterations is set to 5 and the number of Nystrom centers is equal to the number of points in the training set) and we used it both for training and test.

## Modeling the expected performance of individual features

The importance of a feature is quantified by the corresponding optimal test error

$$\frac{\int_0^\infty dx \int_0^R dy\, (y - f^*(x))^2 p(x,y)}{\int_0^R (y - \bar{y})^2 p(y) dy}. \tag{5}$$

where $R$ is the length of the cone. The target, or regression, function is given by $f^*$:

$$f^*(x) = \langle y|x \rangle = \int_0^R y p(y|x) dy \tag{6}$$

and can be shown to minimize the expected error among all possible functions. In practice, the optimal expected error cannot be computed exactly, since neither the target nor the data distribution are known. In this paper, we choose to estimate the regression function from data using Kernel ridge regression with a Gaussian kernel. The choice of this latter approach is due to its nonparametric nature, which ensures that any target function $f^*$ can be recovered provided large enough samples, and more generally that accurate estimates can be derived when only finite data are given *Hastie et al., 2001*; *Steinwart, 2002*. Provided with a kernel ridge regression estimator, an approximation to the optimal test error can then be computed on a hold-out set of data.

Next, we are interested into developing a clearer connection between the above statistical approach and quantities with a direct physical meaning. Towards this end, note that the joint, marginal and posterior distributions, given the prior on $y$ and the likelihood $p(x|y)$ are given by,

$$p(x, y) = p(x|y)p(y) \tag{7}$$

$$p(x) = \int_0^\infty p(x, y) dy \tag{8}$$

$$p(y|x) = \frac{p(x, y)}{p(x)}. \tag{9}$$

Assuming that points are sampled uniformly within the cone of detection, the prior is $p(y) = 2y/R^2$ (thus $\bar{y} = 2R/3$ and the denominator in *Equation 5* is $R^2/18$). Importantly, the likelihood $p(x|y)$ is dictated by the fluid dynamics of odor plumes from a concentrated source. Our features are sample averages of intensity or timing attributes of the odor signal, and in the limit of large samples, they are normally distributed, that is

$$p(x|y) = \mathcal{N}(x - g(y), s(y)). \tag{10}$$

Assuming the asymptotic limit to be well approximated, we can find empirical estimates for $g(y)$ and $s(y)$ given the data. This provides the desired characterization of the expected error and hence of the importance of each given feature in term of predictive power. Indeed, by numerically solving *Equations 5–9* with the assumption (*Equation 10*) and using the empirical estimates for $g(y)$ and $s(y)$, we reproduce the predictive power of individual features showed in *Figure 5c* (see *Figure 5—figure supplement 3*). To move beyond empirical estimates of the likelihood and generalize predictions to other kinds of flows one could generalize asymptotic models of turbulent plumes developed in *Celani et al., 2014* to account for z-variations of the sampling locations. Similarly, extension are needed to consider combination of possibly non Gaussian features.

## Data availability

Simulations of odor transport are generated through NEK5000 *Fischer et al., 2008*, freely available from Argonne National Laboratory (https://nek5000.mcs.anl.gov/). Outputs from DNS presented in *Figure 1* are processed to extract time series and compute the five features described in the text (average concentration, slope, blank duration, whiff duration and intermittency factor). These data are available at the online repository https://osf.io/ja9xr/. The results of kernel ridge regression perfomed on these data are presented in *Figures 2–6*. We perform kernel ridge regression on these data with the freely available code FALKON *Rudi et al., 2018* (https://github.com/LCSL/FALKON_paper, *Rigolli, 2022* copy archived at swh:1:rev:480741cf1e7da0d1d7415309cd6f254080a6ca17).

## Acknowledgements

This work was supported by: the European Research Council (ERC) under the European Union's Horizon 2020 research and innovation programme (grant agreement No 101002724 RIDING); the Air Force Office of Scientific Research under award number FA8655-20-1-7028; the National Institutes of Health (NIH) under award number R01DC018789; the French government, through the UCAJEDI Investments in the Future project managed by the National Research Agency (ANR) under reference number #ANR-15-IDEX-01. The authors are grateful to the OPAL infrastructure from Université Côte d'Azur and the Université Côte d'Azur's Center for High-Performance Computing for providing resources and support. N.M. and N.R. are thankful for the support of Instituto Nazionale di Fisica Nucleare (INFN) Scientific Initiative SFT: Statistical Field Theory, Low-Dimensional Systems, Integrable Models and Applications.

## Additional information

### Competing interests

Agnese Seminara: Reviewing editor, *eLife*. The other authors declare that no competing interests exist.

### Funding

| Funder | Grant reference number | Author |
| --- | --- | --- |
| European Research Council | 101002724 | Agnese Seminara |
| Air Force Office of Scientific Research | FA8655-20-1-7028 | Lorenzo Rosasco |
| National Institutes of Health | R01DC018789 | Agnese Seminara |
| Agence Nationale de la Recherche | ANR-15-IDEX-01 | Agnese Seminara |

The funders had no role in study design, data collection and interpretation, or the decision to submit the work for publication.

## Author contributions
Nicola Rigolli, Conceptualization, Software, Formal analysis, Validation, Investigation, Visualization, Methodology, Writing – original draft, Writing – review and editing; Nicodemo Magnoli, Supervision, Investigation, Project administration, Writing – review and editing; Lorenzo Rosasco, Software, Supervision, Funding acquisition, Methodology, Project administration, Writing – review and editing; Agnese Seminara, Conceptualization, Software, Formal analysis, Supervision, Funding acquisition, Investigation, Visualization, Methodology, Writing – original draft, Project administration, Writing – review and editing

## Author ORCIDs
Nicola Rigolli ![ORCID] http://orcid.org/0000-0002-0734-2105
Agnese Seminara ![ORCID] http://orcid.org/0000-0001-5633-8180

## Decision letter and Author response
Decision letter https://doi.org/10.7554/eLife.72196.sa1
Author response https://doi.org/10.7554/eLife.72196.sa2

# Additional files

## Supplementary files
• MDAR checklist

## Data availability
Simulations of odor transport are generated through NEK5000 (43), freely available from Argonne National Laboratory (https://nek5000.mcs.anl.gov/). Outputs from DNS presented in Figure 1 are processed to extract time series and compute the five features described in the text (average concentration, slope, blank duration, whiff duration and intermittency factor). These data are available at the online repository https://osf.io/ja9xr/. The results of kernel ridge regression performed on these data are presented in Figures 2-6. We perform kernel ridge regression on these data with the freely available code FALKON (https://github.com/LCSL/FALKON_paper, copy archived at swh:1:rev:480741cf1e7da0d1d7415309cd6f254080a6ca17).

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
