## [Editor Report]

This paper explores the question of the optimum strategy for odor detection in a turbulent environment. The authors use high-resolution simulations of turbulent flow to investigate the transport and detection of odors advected by the flow, comparing machine learning strategies based on the temporal dynamics of the signal with those based on intensity. The work should be of interest to researchers working on a broad range of problems in sensation and navigation across scales.

---

## [Decision Letter]

**Decision letter after peer review:**

Thank you for submitting your article "Learning to predict target location with turbulent odor plumes" for consideration by *eLife*. Your article has been reviewed by 2 peer reviewers, one of whom is a member of our Board of Reviewing Editors, and the evaluation has been overseen by Aleksandra Walczak as the Senior Editor. The reviewers have opted to remain anonymous.

Essential revisions:

1) It would be good if the authors could provide evidence that their conclusions do not depend on the precise location of the odor source relative to the cylinder. We do not require an exhaustive study, but some evidence would be very helpful.

2) We think that more could have been understood from these data, had the authors tried to focus on dimensionless quantities. It would be important to understand what sets the distance from the source where the intensity-based search strategy becomes less effective than the time-based one, and how the results depend on the Schmidt number.

3) The authors should carefully specify how dimensionless parameters are introduced. They do it in places but not systematically. For instance, how are wavenumbers in Figure 1(d) are made dimensionless?

4) On a similar note, how does the range of k's in Figure 1(d) compare to the height of the channel? Are these k's taken along the streamwise direction only? What is the vertical axis in that graph? Would it integral over all k's correspond to the odour density variance? These need to be specified.

5) What is the value of odour diffusivity \kapppa_\theta? What is the Schmidt number in these simulations? Visual inspection of Figure 1(d) suggests that Sc>>1, which might explain why the -5/3 slope is so far from representing the data. A reference to passive scalar advection by turbulence is in order here. We would suggest K. Sreenivasan, Turbulent mixing: A perspective, PNAS (2019).

6) Perhaps T in line 335 should be replaced by \theta.

7) Please provide figure numbers in lines 377, 412, and 414.*Reviewer #1 (Recommendations for the authors):*

In this work, the authors combine high resolution numerical simulations of turbulent airflow that advects a passive scalar (an "odor") and machine learning algorithms to investigate the question of how a navigation strategy based on the temporal dynamics of odor detection compares with one based on the intensity of the plume. In the simulations, the odor is introduced downstream of a cylinder that induces turbulence in the flow and the machine learning algorithm is trained and evaluated at various points further downstream. The authors conclude that intensity/gradient based measurements work closer to the source, while temporal schemes are good throughout the range. The study is done to a very high standard and presented clearly, although there are general questions about the data analysis (see below) that should be improved. That said, the work should have significant impact on a broad range of fields, from sensing to navigation, across a range of organism length scales.

One concern is the nature of the turbulent profile and its advection of the passive scalar. It would help if the characteristic dimensionless numbers of the problem were specified (Peclet, Schmidt numbers), and it was made clear how the choice of odor release point affects the conclusions. The turbulence itself spreads and diffuses with distance from the source, and one would presume the location of the odor release can matter substantially.

Also, as mentioned briefly in the Discussion, this work examines algorithms based on evaluating accuracy of prediction at a given measurement point. Unless I have missed something in the presentation, the issue of navigation is left unexamined. As in bacterial chemotaxis or any related problem, no navigation strategy is perfect, especially in the face of such a fluctuating source of information, so the full problem involves making estimates of where the source is and then moving to a new location, estimating again, moving, etc. The authors should clarify under what circumstances an evaluation at a fixed point is sufficiently predictive of "learning".*Reviewer #2 (Recommendations for the authors):*

In their manuscript, Rigolli et al., studied how measurements of intensity of a passive scalar (odour), its spatial gradients, and its time variations can be used to efficiently find the spatial location of its source. To this end, the authors performed direct numerical simulations of high-Reynolds number pressure-driven channel flow, partially blocked by a cylinder to generate turbulent velocity fluctuations. At a fixed position downstream the cylinder, they introduce a source of odour that diffuses through the fluid and is advected by its turbulent motion. The authors then trained a supervised machine learning algorithm on a collection of spatial and temporal odour profiles thus obtained. The algorithm was subsequently tested on various odour signals originating from the same pool of simulations. By varying the spatial position of the measurement point and the temporal exposure, the authors concluded that detection strategies based on the odour concentration and its spatial gradient work best at small separations from the source (and also close to the zero-odour concentration boundaries), while time-based strategies work reasonably well everywhere within the domain selected for detection.

Within the geometry set by the authors, the conclusions of the paper are supported by data. However, this setup leads to a natural question whether the search strategies determined here pertain to the distance to the odour source or to the distance to the turbulence source (the half-cylinder). Turbulence generated by an obstacle has a particular spatial profile; its temporal profile also depends on the distance to the obstacle. It is therefore possible that the machine learning algorithm has indirectly picked up these features rather than the odour profile itself. This can be settled by feeding the existing algorithm signals from simulations with various distances between the source and the obstacle. In the absence of this check, it is impossible to make general statements about applicability of these search strategies in other situations.

Another weak point of this study is the use of a cone of detection as it dramatically reduces the search complexity to almost a quasi one-dimensional problem. The authors appreciate it and point out that their choice is forced by a very limited by a very small number of detections outside the cone. This shortcoming should be addressed in future work.

---

## [Author Response]

Essential revisions:1) It would be good if the authors could provide evidence that their conclusions do not depend on the precise location of the odor source relative to the cylinder. We do not require an exhaustive study, but some evidence would be very helpful.

We expect results to be robust to changes in source location, as long as this is within the bulk of the channel and not within the viscous layer, which generates substantially less intermittent odor plumes (see e.g. Fackrell and Robins, J Fluid Mech 117:1, 1982). We performed additional computational fluid dynamics simulations, moving the source to two further locations downstream of the obstacle, at the same height and performed inference over an identical conical domain shifted to match the source location (see Figure 2, Figure Supplement 5 left). Performance at source height varies little over the three locations, demonstrating that the algorithm learns from the dynamics of the scalar and not of the underlying velocity field (we chose one individual feature per class and their combination, see Figure 2, Figure Supplement 5 center). For all three sources, performance of the average degrades with height, whereas performance of intermittency improves with height (Figure 2, Figure Supplement 5 right), consistent with the results of the Figure 5c and despite using a different cone. Finally, we trained the algorithm with a dataset generated by one source and tested it on dataset obtained from the other sources: we found that pairs of features are more sensitive to the details of the dataset, whereas individual features maintain their full predictive power (Figure 2 Figure Suppl 6).

If and how animals adapt their models to deal with sources at different heights is a fascinating question that is relevant for animal behavior; we believe this point deserves an in depth investigation on its own and we are currently addressing these questions in a different study.

**Author response image 1. sa2fig1:** Estimated test error of individual features at different heights using the theoretical framework Equation (1)-(5) outlined in the text with the empirical likelihood estimated from data (not shown). Symbols as in Figure 5c of the manuscript (black / grey represent timing /intensity features; dashed lines: whiffs).

2) We think that more could have been understood from these data, had the authors tried to focus on dimensionless quantities. It would be important to understand what sets the distance from the source where the intensity-based search strategy becomes less effective than the time-based one, and how the results depend on the Schmidt number.

We thank the reviewers for this important observation that led us to develop a theoretical frame work that we believe deserves further attention. We highlight below the conceptual building blocks, which we have detailed in the manuscript. The predictive power χ of an individual feature x is its expected error: ∫0R(y−f∗(x))2p(x,y)dy∫0R(y−y¯)2p(y)dy where *p*(*x,y*) is the joint probability distribution of the input output pair (*x,y*)and *p*(*y*) is the prior on the output *y.f*(x)* is the optimal predictor and can be shown to be f∗(x)=∫0Ryp(y|x)dy, where R is the length of the cone (see Hastie et al., The elements of statistical learning: Datamining, inference, and prediction, 2001 and Steinwart, J of Complexity 18:768, 2002). The expected error is generally unavailable because p<milestone-start />(<milestone-end />yx<milestone-start />)<milestone-end /> and p(x,y) are unknown. However, we can connect equation (1) to the physics of the odor plume by computing the joint, marginal and posterior distributions using the likelihood p(x│y):p(x,y)=p(x|y)p(y)
p(x)=∫0∞p(x,y)dy
p(y|x)=p(x,y)p(x) where the prior is *p(y)*=2y/*R^2^* (thus y¯= 2*R*/3 and the denominator in equation (1) is R^2^/18). Importantly, the likelihood is dictated by the fluid dynamics of odor plumes from a concentrated source. Because our individual features are sample averages, their statistics is approximately Gaussian: p(x|y)−N(x−g(y),s(y)) and we can then simply characterize the likelihood by data fitting to estimate their average and standard deviation. By solving equations (1) to (4) with the assumption (5) and the empirical estimates for g(y) and s(y) we recover the predictive power of individual features showed in Figure 5c of the manuscript (see Figure 1). (The argument needs to be better adapted to whiffs which deviate considerably from a normal distribution)

We believe this theoretical framework deserves further attention. Indeed, one can move

beyond empirical estimates of the likelihood and generalize predictions to pairs of features and other kinds of flows by leveraging the asymptotic arguments recently proposed by Celani et al., Phys Rev 4:041015, 2014. We do not further elaborate on this point here, but we are currently working to fully develop these asymptotic arguments not only to obtain scaling behaviors for g(y) and s(y) but also the prefactor, which is crucial to understand what non-dimensional quantities dictate the switch from timing to intensity. Note that in this regime the Schmidt number plays a minor role, as we further elaborate on below.

3) The authors should carefully specify how dimensionless parameters are introduced. They do it in places but not systematically. For instance, how are wavenumbers in Figure 1(d) are made dimensionless?

Thank you for the comment, we have provided information on the Schmidt number used in the simulations (Sc = 1) and the definition of wavenumbers in Figure 1d (see next answer for more details on this point), as well as dimensional parameters that were either only mentioned in the text (viscosity) or not mentioned (diffusivity). All information is now summarized in Table 1.

4) On a similar note, how does the range of k’s in Figure 1(d) compare to the height of the channel? Are these k’s taken along the streamwise direction only? What is the vertical axis in that graph? Would it integral over all k’s correspond to the odour density variance? These need to be specified.

Thank you for the comment, we have included the missing information. Figure 1d shows the two dimensional spectra Ek=ddk(∫│kxy│<k│c^(kxy)│2d2k) normalized with the scalar variance *σ*c2, where ĉ(κ_xy_) is the two dimensional Fourier transform of the scalar concentration at the height of the source. The integral of the spectra is indeed the scalar variance. We non-dimensionalize the wavenumber with the inverse Kolmogorov scale *η* −1. We corrected a small discrepancy coming from the fact that the Legendre polynomials used by Nek5000 have sampling points that are not perfectly equally spaced. The k−5∕3 scaling holds for k *η* 0.1, consistent with previous experimental results in channel flow (see e.g. S.G. Saddoughi and S.V. Veeravalli J. Fluid Mech.(1994) 268:333-372). To further characterize the flow we add a panel to figure 1, showing that the mean flow follows the well known law of the wall, recovering classical statistics for channel turbulence.

5) What is the value of odour diffusivity \kapppa_\theta? What is the Schmidt number in these simulations? Visual inspection of Figure 1(d) suggests that Sc>>1, which might explain why the -5/3 slope is so far from representing the data. A reference to passive scalar advection by turbulence is in order here. We would suggest K. Sreenivasan, Turbulent mixing: A perspective, PNAS (2019).

Here we work at Schmidt = 1, κ*θ* = ν = 1.5 × 10−5m2∕s. The inertial range is short but consistent with previous literature on channel flow (see previous answer). Varying the Schmidt number affects the fine details between the Batchelor and Kolmogorov scales (Falkovich et al., Rev Mod Phys 105 73:913, 2001). These are below the size of the source in our simulations, as is also the case in 106 many situations of practical relevance. In this regime, the dynamics below the Kolmogorov scale 107 affects little the large scale statistics of odor plumes from concentrated sources which are dictated 108 by the separation of Lagrangian particles (as argued in Celani et al., Phys Rev X, 2014 and confirmed experimentally in Duplat et al., Phys Fluids, 22:035104, 2010). We included this comment at page 12.

6) Perhaps T in line 335 should be replaced by \theta.

Thank you, this is indeed a typo and should read c

7) Please provide figure numbers in lines 377, 412, and 414.

Thank you, we have added the references to the figures

Reviewer #1 (Recommendations for the authors):In this work, the authors combine high resolution numerical simulations of turbulent airflow that advects a passive scalar (an "odor") and machine learning algorithms to investigate the question of how a navigation strategy based on the temporal dynamics of odor detection compares with one based on the intensity of the plume. In the simulations, the odor is introduced downstream of a cylinder that induces turbulence in the flow and the machine learning algorithm is trained and evaluated at various points further downstream. The authors conclude that intensity/gradient based measurements work closer to the source, while temporal schemes are good throughout the range. The study is done to a very high standard and presented clearly, although there are general questions about the data analysis (see below) that should be improved. That said, the work should have significant impact on a broad range of fields, from sensing to navigation, across a range of organism length scales.One concern is the nature of the turbulent profile and its advection of the passive scalar. It would help if the characteristic dimensionless numbers of the problem were specified (Peclet, Schmidt numbers), and it was made clear how the choice of odor release point affects the conclusions. The turbulence itself spreads and diffuses with distance from the source, and one would presume the location of the odor release can matter substantially.

We thank the reviewer for the comment. We have now stated the non-dimensional numbers and remarked that in our regime the Schmidt number is expected to play a minor role (see response to general comments above). To analyze how the results depend on source location, we have conducted a series of numerical simulations with the source located further downstream from its original location, and this does not affect our conclusion. We did not move the source laterally, as this cannot affect the results because the flow is homogeneous in the crosswind direction z2. We expect that moving the source to the ground will affect the results because intermittency of an odor plume from a source located at ground is much less intermittent (see response to general comment 1). Animals that would target both sources in air and on ground would clearly have to use qualitatively different predictive models. Ethological studies on dogs suggest that they do track both sources on ground and in air, and that they do this by alternating sniffing in air and on ground.

We are currently investigating this fascinating behavior.

Also, as mentioned briefly in the Discussion, this work examines algorithms based on evaluating accuracy of prediction at a given measurement point. Unless I have missed something in the presentation, the issue of navigation is left unexamined. As in bacterial chemotaxis or any related problem, no navigation strategy is perfect, especially in the face of such a fluctuating source of information, so the full problem involves making estimates of where the source is and then moving to a new location, estimating again, moving, etc. The authors should clarify under what circumstances an evaluation at a fixed point is sufficiently predictive of "learning".

We agree entirely with the referee on this point, which we now stress more in the Discussion. We are currently working to test whether the best predictors are also the best features to be used in navigation. Although it is often implicitly assumed that this is indeed the case, up until now this question is entirely unexplored.

Reviewer #2 (Recommendations for the authors):In their manuscript, Rigolli et al., studied how measurements of intensity of a passive scalar (odour), its spatial gradients, and its time variations can be used to efficiently find the spatial location of its source. To this end, the authors performed direct numerical simulations of high-Reynolds number pressure-driven channel flow, partially blocked by a cylinder to generate turbulent velocity fluctuations. At a fixed position downstream the cylinder, they introduce a source of odour that diffuses through the fluid and is advected by its turbulent motion. The authors then trained a supervised machine learning algorithm on a collection of spatial and temporal odour profiles thus obtained. The algorithm was subsequently tested on various odour signals originating from the same pool of simulations. By varying the spatial position of the measurement point and the temporal exposure, the authors concluded that detection strategies based on the odour concentration and its spatial gradient work best at small separations from the source (and also close to the zero-odour concentration boundaries), while time-based strategies work reasonably well everywhere within the domain selected for detection.Within the geometry set by the authors, the conclusions of the paper are supported by data. However, this setup leads to a natural question whether the search strategies determined here pertain to the distance to the odour source or to the distance to the turbulence source (the half-cylinder). Turbulence generated by an obstacle has a particular spatial profile; its temporal profile also depends on the distance to the obstacle. It is therefore possible that the machine learning algorithm has indirectly picked up these features rather than the odour profile itself. This can be settled by feeding the existing algorithm signals from simulations with various distances between the source and the obstacle. In the absence of this check, it is impossible to make general statements about applicability of these search strategies in other situations.

Thank you for your comment. To test robustness of the algorithm, we have performed two additional sets of simulations moving the source further downstream of the obstacle: the results are consistent with the results of the main simulation; source location may affect details about where exactly the transition between the timing vs intensity regimes occurs. First, we verified that performance of 2 individual features and their pair does not depend on source location, confirming that the algorithm learns solely from odor statistics. Second, we trained the algorithm with data from one simulation and tested its performance over datasets from the remaining simulations.

We found that performance of individual features is preserved even when training and test are performed over different dataset. Pairs of features are more sensitive to consistency between training and test dataset (see Figure 2 Figure supplement 6). Third, we analysed performance for each source location as a function of height of the sampling plane. We recovered that timing features improve with height whereas intensity features degrade with height, as showed in Figure 5c (Figure 2 Figure Supplement 5, right). We note that the transition between timing vs intensity may occur at different heights for the three source locations, we did not investigate this comparison in more detail.

Another weak point of this study is the use of a cone of detection as it dramatically reduces the search complexity to almost a quasi one-dimensional problem. The authors appreciate it and point out that their choice is forced by a very limited by a very small number of detections outside the cone. This shortcoming should be addressed in future work.

We entirely agree with the referee, in fact olfactory searches first need to locate the plume and subsequently locate the source within the plume. Our manuscript defines the most informative features once the agent is within the plume. We believe that our results are not relevant for the previous stage when the agent searches for the plume. This is because there is hardly any detection at all outside of the cone. In fact, our ongoing work (currently under review) demonstrates a model based navigation strategy that uses absence of detection to narrow down the possible locations to search for the plume.